# Identification of the Immune Landscapes and Follicular Helper T Cell-Related Genes for the Diagnosis of Age-Related Macular Degeneration

**DOI:** 10.3390/diagnostics13172732

**Published:** 2023-08-22

**Authors:** Yao Yang, Zhiqiang Sun, Zhenping Li, Que Wang, Mingjing Yan, Wenlin Li, Kun Xu, Tao Shen

**Affiliations:** 1The Key Laboratory of Geriatrics, Beijing Institute of Geriatrics, Institute of Geriatric Medicine, Chinese Academy of Medical Sciences, Beijing Hospital/National Center of Gerontology of National Health Commission, Beijing 100730, China; 2Department of Neurosurgery, Renmin Hospital of Wuhan University, Wuhan 430060, China; 3Central Laboratory, Renmin Hospital of Wuhan University, Wuhan 430060, China; 4Department of Ophthalmology, Guangdong Provincial Key Laboratory of Malignant Tumor Epigenetics and Gene Regulation, Sun Yat-Sen Memorial Hospital, Sun Yat-Sen University, Guangzhou 510120, China

**Keywords:** follicular helper T cells, age-related macular degeneration, diagnosis, biomarkers, bioinformatics

## Abstract

Background: Age-related macular degeneration (AMD) is a progressive ocular ailment causing age-associated vision deterioration, characterized by dysregulated immune cell activity. Notably, follicular helper T (Tfh) cells have emerged as pivotal contributors to AMD pathogenesis. Nonetheless, investigations into Tfh-associated gene biomarkers for this disorder remain limited. Methods: Utilizing gene expression data pertinent to AMD procured from the Gene Expression Omnibus (GEO) repository, we employed the “DESeq2” R software package to standardize and preprocess expression levels. Concurrently, CIBERSORT analysis was utilized to compute the infiltration proportions of 22 distinct immune cell types. Subsequent to weighted gene correlation network analysis (WGCNA), coupled with differential expression scrutiny, we pinpointed genes intricately linked with Tfh cells. These potential genes underwent further screening using the MCODE function within Cytoscape software. Ultimately, a judicious selection of pivotal genes from these identified clusters was executed through the LASSO algorithm. Subsequently, a diagnostic nomogram was devised based on these selected genes. Results: Evident Tfh cell disparities between AMD and control cohorts were observed. Our amalgamated analysis, amalgamating differential expression data with co-expression patterns, unveiled six genes closely associated with Tfh cells in AMD. Subsequent employment of the LASSO algo-rithm facilitated identification of the most pertinent genes conducive to predictive modeling. From these, *GABRB3*, *MFF*, and *PROX1* were elected as prospective diagnostic biomarkers for AMD. Conclusions: This investigation discerned three novel biomarker genes, linked to inflammatory mechanisms and pivotal in diagnosing AMD. Further exploration of these genes holds potential to foster novel therapeutic modalities and augment comprehension of AMD’s disease trajectory.

## 1. Introduction

Age-related macular degeneration (AMD) is a primary cause of irreversible blindness in the elderly. A meta-analysis predicts that AMD cases could reach 288 million by 2040 [1]. The disease’s origin involves lipid metabolism, oxidative stress, toxicity, aging, and genetics [2]. Among these factors, aging, with its systemic inflammation, prominently emerges as AMD’s primary risk. Inflammation relates to AMD susceptibility genes that encode complement factors, drusen-based complement proteins, and immune activation features, including inflammasome activation. The retinal pigment epithelium (RPE) is responsible for phagocytizing waste, like lipofuscin from photoreceptor outer segments, and releasing it into the choroidal circulation through Bruch’s membrane. Aging reduces RPE’s capacity to clear lipofuscin, leading to intracellular buildup [3]. This impairs RPE’s phagocytic function, causing abnormal debris, lipid, and immune molecule deposition between RPE and Bruch’s membrane, culminating in drusen formation. Drusen, AMD’s hallmark, potentially links to complement activation. Subsequent activation harms choroidal capillary endothelial cells and RPE, contributing to AMD onset. These findings underscore inflammation’s pivotal role in AMD’s pathogenesis [4]. Thus, we investigate AMD and inflammation’s intricate interplay.

AMD causes damage to the photoreceptor layer, RPE, Bruch’s membrane, and choroidal capillary layer within the macula, leading to vision loss. AMD is categorized into early, intermediate, and advanced non-neovascular AMD (dry AMD) or advanced neovascular AMD (wet AMD). Dry AMD can serve as a precursor to wet AMD, as retinal dysfunction causes atrophic changes, subsequently progressing to wet AMD [5]. Extracellular deposits within drusen in dry AMD induce RPE ischemia, prompting the release of angiogenic factors by RPE cells, including vascular endothelial growth factor (VEGF). This cascade gives rise to choroidal neovascularization, contributing to the development of wet AMD [6]. Wet AMD can cause significant visual impairment and can be treated with laser therapy and intravitreal anti-VEGF injections, but this imposes medical burdens. Dry AMD only has complement-based treatments as potentially effective options, with uncertain efficacy and a likelihood of progression to wet AMD. Regardless of the type of AMD and its stage of progression, it places a significant medical burden on society [7]. Therefore, the early diagnosis of AMD is of great significance, which is helpful for the prevention and early intervention of AMD. This study focuses on the immune mechanism of AMD pathogenesis and its potential for early diagnosis.

T cells, a type of immune cell, play a pivotal role in cellular immunity. Based on their functions, T cells can be categorized into various subtypes, including helper T cells (Th cells), inhibitory T cells (Ts cells), cytotoxic T cells (CTL or Tc cells), and delayed type hypersensitivity T cells (TDTH cells). T cells can also be classified into CD4+ and CD8+ subgroups based on the differentiation antigens on their cell surface. Consequently, other T cell types or functions might also be involved in the pathogenesis of AMD [8,9]. Follicular helper T (Tfh) cells stem from CD4+ T cells and express the C-X-C chemokine receptor type 5. They secrete IL-4, IL-10, and IL-21, thereby stimulating the differentiation, proliferation, and maturation of B cells [10]. Tfh cells can maintain this phenotype through continuous expression of B-cell CLL/lymphoma 6 during interactions with homologous B cells at the B-T boundary [11]. Tfh cells have associations with various diseases, including primary immunodeficiency, hyperimmunoglobulin M syndrome, autoimmune myasthenia gravis, autoimmune thyroid disease, systemic lupus erythematosus, Sjogren’s syndrome, and others [12]. Besides safeguarding against infectious diseases, Tfh cells also contribute to the pathogenesis of autoimmune diseases, atherosclerosis, and various cancers [13].

The precise role of Tfh cells in immunity remains not fully elucidated, and their implication in AMD’s pathogenesis is yet to be definitively established. There is a requirement to leverage high-throughput techniques and conduct bioinformatics analyses to gain a more profound understanding of the underlying genetic and molecular mechanisms governing this disease. This approach could potentially offer insights into harnessing Tfh cells as diagnostic biomarkers for AMD and identifying pivotal genes implicated in the disease process, thereby potentially leading to novel treatment strategies.

## 2. Materials and Methods

### 2.1. Data Acquisition and Filtering

For this study, we obtained the GSE29801 dataset as the training set and the GSE99248 dataset as the testing set from the Gene Expression Omnibus (GEO) website (http://www.ncbi.nlm.nih.gov/geo/, accessed on 10 January 2023). The GSE29801 dataset contains gene expression data and relevant clinical information from macular and extramacular RPE-choroid and retina tissues of both AMD patients and normal donors [14]. To ensure accurate data interpretation, a total of 151 normal samples and 142 AMD samples were included in the analysis of GSE29801. The GSE99248 dataset contains RPE tissues from sixteen AMD patients and fifteen normal patients [15]. To normalize the expression file and enhance data accuracy, we utilized the “vst” function within the “DESeq2” R package [16].

### 2.2. Gene Set Variation Analysis

Gene Set Variation Analysis (GSVA) was used to investigate potential biological mechanisms involved in the occurrence and progression of AMD by comparing gene expression between AMD and normal groups. For annotation purposes, we downloaded the “h.all.v2022.1.Hs.symbols.gmt” file from the GSEA website (http://www.gsea-msigdb.org/gsea/index.jsp, accessed on 21 January 2023) [17]. Next, to identify differential gene sets between the two groups, a Student’s *t*-test was conducted and significant gene sets were considered at *p*-value < 0.05.

### 2.3. CIBERSORT Analysis

To calculate the infiltration levels of 22 immune cells in each sample, we employed the CIBERSORT method of the IBOR package [18]. In the calculated files, samples with a *p*-value < 0.05 were considered significant and selected for further analysis, while those that did not meet this criterion were eliminated. The relationships between all immune cells were evaluated using the “ggcorrplot” package in R software to generate a correlation heatmap. Additionally, the “scatterplot3d” package was used to conduct Principal Component Analysis (PCA).

### 2.4. Weighted Gene Correlation Network Analysis

By performing weighted gene correlation network analysis (WGCNA) [19], we constructed a correlation heatmap between the genes of AMD patients and four types of immune cells. To identify the best soft-power, we utilized the scale independence and mean connectivity method, with a value of 17 being determined as the optimal soft-power. The similar module clustering method was then conducted to determine the best cutoff value, which was calculated as 0.4. A minimum module size of 30 was set to remove genes with low expression levels and cluster the filtered genes into several distinct modules. Hub genes were identified based on their correlation levels (>0.8) between genes and modules and gene significance (>0.76). Furthermore, we calculated the correlations between modules and the four types of immune cells, and a *p*-value < 0.05 was considered significant. The Gene Ontology (GO) analysis was performed in R software (Version 3.4.1) to investigate the biological potentials of these identified hub genes, with an adjusted *p*-value and *q*-value both set at <0.05.

### 2.5. Differently Expressed Genes Analysis

In the first step of our analysis, we conducted differential expression analysis using the “DESeq2” package to identify genes that were differentially expressed between AMD and normal samples. Genes with an absolute value of *logFC* > 1 and adjusted *p*-value < 0.05 were considered as differently expressed genes (DEGs). The Venn diagram was then used to show the intersected genes between the DEGs and WGCNA module. Next, we used the STRING database (https://cn.string-db.org/, accessed on 11 January 2023) to analyze the relationships between these intersected genes. The MCODE function in Cytoscape software (Version 2.1) was then used to identify the hub genes among these intersected genes. The “ggplot2” package was employed to visualize the relative expression levels of these hub genes.

### 2.6. Identified the Potential Immune-Related Biomarkers of AMD

Firstly, we randomly divided 183 samples into a training group and a test group using R software. We then performed lasso regression analysis and multivariate analysis on the extracted hub genes within the training group to identify diagnostic markers using the “glmnet” package. Next, we constructed a Nomogram for predicting the risk of AMD and drew its calibration curve using the “Hmisc” package. We also compared the diagnostic value of our model, age, and gender by drawing ROC curves using the “ROCR” package. The risk scores for each sample were obtained using the “predict” function in the “rms” package. Finally, we verified the correlation between calculated risk scores and Tfh cell infiltration levels in each group using Spearman correlation analysis.

## 3. Results

### 3.1. Study Process and Data Normalization

The flowchart in Figure 1 depicts the study process. By integrating the results from GSVA and CIBERSORT, we identified several immune cells, particularly Tfh cells, that were crucial in the development of AMD. Additionally, the WGCNA and differential expression analysis helped us identify six Tfh-related genes that could serve as potential biomarkers for AMD.

AMD is a chronic and progressive disease. Early-stage AMD can advance to late-stage AMD, and dry AMD can progress to wet AMD under certain conditions. Therefore, it would be beneficial to include various types of AMD in studies examining the chronic progression of the disease. We utilized GSE29801, which included all types of AMD, as the raw data for this study, annotated by the GPL4133 platform (Agilent-014850 Whole Human Genome Microarray 4 × 44 K G4112F), which consisted of 151 normal tissue samples and 142 AMD tissue samples. After normalization and removal of batch effects, as shown in Figure 2A,B, we selected 293 samples (151 normal retinas and 142 AMDs) for further analysis, with 19,749 genes available for subsequent examination. Through the use of GSVA, we identified upregulated biological functions related to immune activities in AMDs, including IL6-JAK-STAT3-signaling, interferon (IFN)-α response, interferon (IFN)-γ response, inflammatory response, and TNF-α signaling via NF-kB (Figure 2C). These results further support the significant relationship between AMD and immune processes.

### 3.2. Infiltration Levels of Immune Cells

The CIBERSORT method in R software was used to estimate the infiltration levels of 22 immune cell types in all samples. Samples with *p*-values ≥ 0.05 were removed, resulting in 183 samples for further analysis, as shown in Figure 3A. The correlation heatmap in Figure 3B revealed the correlation values of the 22 immune cell types, showing a positive relationship between Tfh cells and regulatory T cells and a negative relationship between Tfh cells and monocytes. In Figure 3C, the boxplot exhibited the relative infiltration levels of immune cells between AMD and normal groups, showing that the levels of plasma cells, Tfh cells, Tregs, activated mast cells, M0 macrophages, M1 macrophages, and M2 macrophages were higher in AMD groups than normal groups, while the levels of CD8^+^ T cells, activated NK cells, and monocytes were lower. Additionally, the PCA plot showed a significant difference between AMD and normal samples based on the infiltration levels of immune cells (Figure 3D).

### 3.3. Filtration of Immune Cell-Related Modules

To identify significant gene modules among genes and four immune cells (M1 macrophages, Tregs, Tfh cells, and CD8+ T cells), we used WGCNA on 95 AMD samples. We first constructed scale independence and mean connectivity plots (Figure 4A) and chose a soft-power of 17. Hierarchical clustering was then used to show gene distributions among different modules, with different colors representing different modules (Figure 4B). We found that there were 11 modules among all genes (Figure 4C) and that the salmon module had the highest correlation value of 0.85 and the lowest *p*-value of 9 × 10^−27^, making it the hub module. In the salmon module of Tfh cells, we identified 782 significant genes with module membership > 0.8 and gene significance > 0.76 (Appendix A). We found a strong correlation between the 782 significant genes and Tfh cells, with a correlation coefficient of 0.91 and a *p*-value of less than 1 × 10^−200^ (Figure 4D). Based on these genes, we conducted a GO analysis and found numerous enriched terms in the biological processes of the eye, such as photoreceptor cell maintenance, detection of visible light, retina development in camera-type eye, eye development, and visual system development (Figure 4E). These results provide further evidence that hub genes related to Tfh cells play a crucial role in the visual impairments associated with AMD progression [20].

### 3.4. Identified the Tfh-Related Hub Genes

We conducted a thorough analysis of gene expression in 183 samples to uncover potential genetic factors involved in AMD development. Our analysis revealed 1291 DEGs (Appendix A), with 560 genes upregulated and 731 downregulated in the AMD cluster compared to the normal cluster (Figure 5A). An overlap analysis was subsequently conducted to identify DEGs that were also related to Tfh biology. As a result, 42 genes (Appendix A) were found to be common between the two sets of genes, providing insight into the potential involvement of Tfh-related mechanisms (Figure 5B) in the development of AMD. The PPI network was exhibited the correlations among the 42 genes (Figure 5C). Moreover, in Figure 5D, *AMACR*, *CD99*, *GABRB3*, *MFF*, *PROX1*, and *RARA* were identified as the hub genes by the MCODE function in Cytoscape software. The bubble chart generated from the GO analysis of the six identified genes showed that they were enriched in the processes of epithelial cell and T-helper cell, which is consistent with the known functions of Tfh cells (Figure 5E). Additionally, the expression levels of these six genes were significantly higher in AMD samples compared to normal samples (Figure 5F). This provides further evidence of their potential role in the development of AMD.

### 3.5. Development of Risk-Model and Construction of the Diagnostic Nomogram

Firstly, we randomly divided 183 samples from GSE29801 into a training (*n* = 92) and testing group (*n* = 91). Within the training group, we used lasso regression to select the five genes with the smoothest *λ*-curve and smallest log(*λ*-value) (*λ* = 5) from among the six hub genes (Figure 6A,B). After performing multivariate analysis on these five genes, we excluded any genes with a *p*-value greater than 0.05. Ultimately, three genes, *GABRB3*, *PROX1,* and *MFF*, were selected (Table 1). We developed a nomogram based on these three genes and two clinical characteristics, age and gender, models to aid in the clinical diagnosis and prediction of AMD (Figure 6C). As shown in Figure 6D, the calibration curve demonstrates that the position between the ideal curve and the apparent curve is extremely close, indicating that our nomogram has an outstanding predictive effect for diagnosing AMD. The AUC values for our model in the training, testing, and GSE99248 group were 0.856, 0.753, and 0.679 respectively (Figure 6E–G). These values were higher than those for clinical characteristics such as age and gender. This demonstrates that our model has significant diagnostic value for AMD when compared to clinical characteristics such as age and gender.

### 3.6. Correlation Analysis between Model-Related Risk Score and Tfh Cells Infiltration Levels

To investigate the relationship between our model’s risk score and Tfh cell infiltration levels, we used the CIBERSORT algorithm to calculate immune cell infiltration levels for each sample in the GSE99248 group. In Figure 7A, the bar-plot presented the infiltration levels of 22 immune cells in 31 samples. A boxplot shows the differences in these 22 immune cells between normal and AMD groups. Among them, Tfh cells and M0 and M2 macrophages were significantly higher in the AMD group than in the normal group (consistent with results from GSE29801), while M1 macrophages were lower in the AMD group (contrary to results from GSE29801) (Figure 7B). Then, we performed Spearman correlation analysis between our model’s genes score and immune cell infiltration levels in the GSE29801and GSE99248 group (Figure 7C,D). The heatmaps revealed that the *R*-values between these three genes and Tfh cells were greater than 0.56, which were the highest compared with other 21 immune cells, in all two cohorts, demonstrating significantly close correlations between them. This suggests that Tfh cells may play an important role in the development of AMD.

## 4. Discussion

The aim of our study was to identify potential AMD biomarkers associated with immune cells. For this purpose, we analyzed 293 tissue samples including 142 AMD patients and 151 controls using the CIBERSORT method to estimate the levels of various immune cells. Our results showed that several types of immune cells, such as plasma cells, Tfh cells, Tregs, activated mast cells, M0 macrophages, M1 macrophages, and M2 macrophages, were more highly infiltrated in AMDs compared to the control group. Through combining the results of WGCNA, differently expressed analysis, and MCODE function in Cytoscape, we identified six Tfh-related hub genes of AMD. By combining the lasso algorithm with multivariate analysis, we identified three AMD diagnostic biomarkers, *GABRB3*, *MFF*, and *PROX1*, and constructed a diagnostic model. We then developed nomograms to improve the clinical application of our diagnostic models. The ROC curve demonstrated that our diagnostic model had better predictive value for AMD compared to clinical characteristics such as age and gender.

As a fundamental component of the retina, the RPE plays an important role in protecting the retina and nerves from UV damage and preventing excessive accumulation of reactive oxygen species [21]. The damaged retina transforms into an inflammatory environment, and as AMD advances, an increasing accumulation of inflammatory molecules occurs, which subsequently leads to inflammation and dysfunction of the RPE. The disease then advances to the next stage known as choroidal neovascularization. In this stage, the absence of RPE and choroidal capillaries contributes to the formation and accumulation of pigments [22].

Tfh cells are a key component of humoral immunity and can be differentiated from CD4+ T cells in germinal centers. Tfh cells interact with B cells in germinal centers and regulate their function, ultimately leading to the differentiation of B cells into plasma cells and memory B cells. Tfh cells provide various signals to B cells during this process, including IL-21 and IL-9 [23]. Early studies have shown that B cells produce a large number of autoantibodies. And the presence of anti-retinal autoantibodies in the serum of patients with dry AMD suggests that B cells play an important role in the pathogenesis of age-related retinopathy [24]. Tfh cells are a unique T cell subpopulation that expresses C-X-C chemokine receptor type 5 on their surface and high levels of B-cell CLL/lymphoma 6. In addition to their helper function for B cells, Tfh cells can also release cytokines such as IFN-γ, IL-17, IL-10, and IL-20, which are directly involved in the inflammatory process [25,26].

However, limited studies have investigated the differences in immune cell infiltration levels during the progression of AMD compared to normal subjects. In this study, we analyzed differences in the proportion of infiltrating immune cells in the retina of AMD patients and normal subjects using the CIBERSORT method. Subsequently, we identified genes related to Tfh cells in AMD samples and found several central genes that exhibited significant statistical differences. The results showed that *GABRB3*, *MFF*, and *PROX1* were highly expressed in samples from AMD patients and were associated with Tfh cells. *GABRB3* is an important neurodevelopmental gene that has been extensively studied in epilepsy and is largely associated with various brain dysplasia [27]. However, there is no research on the role of *GABRB3* in retinal diseases. Our study found that this gene is highly expressed in Tfh cells in AMD samples, and future mechanism studies can focus on the relationship between this gene and the disease. Mitochondrial fission factor (MFF) is a protein that acts on mitochondrial fission by phosphorylation [28]. It has been implicated in various tumours and ocular diseases such as diabetes retinopathy and dry eye [29,30]. Mitochondria are important organelles that provide energy to the cell, and maintaining mitochondrial homeostasis through the processes of fusion and fission is important for mitochondrial function [31]. Additionally, fission of mitochondria also occurs during cell growth, allowing cells to proliferate quickly by providing them with additional energy to the cells [32]. Our research suggests that *MFF* may be highly expressed in Tfh cells in AMD and may contribute to the increased number of Tfh cells in germinal centers, which enhances the proinflammatory responses. Further studies are needed to fully understand the mechanism by which *MFF* regulates mitochondria fission in Tfh cells and its role in AMD. *PROX1* is a gene that plays a critical role in embryonic development and is involved in the progression of various cancers. Its expression and function changes can promote cell proliferation and metastasis in cancer cells through mechanisms such as lymphangiogenesis. In breast cancer, *PROX1* activates the WNT/β-catenin signaling pathway in conjunction with hnRNPK, leading to cancer cell invasion and metastasis. In addition, high expression of *PROX1* in the nucleus of neurocytoma is correlated with an increase in the ratio of undifferentiated to poorly differentiated cells, suggesting a role for *PROX1* in cell proliferation [33]. Tfh cells play a pro-inflammatory role but have low viability compared to other types of T cells. It is hypothesized that the overexpression of genes involved in cell proliferation may increase the number of Tfh cells. This study suggests, for the first time, that *PROX1* is upregulated in Tfh cells from AMD patients, which may lead to the accelerated proliferation of Tfh cells in germinal centers [34].

## 5. Conclusions

In conclusion, our study provides evidence that Tfh cells play an important role in the pathogenesis of AMD by combining CIBERSORT and WGCNA analysis. We identified six Tfh cell-related genes through differently expressed analysis and MCODE function in Cytoscape. Notably, we identified three AMD diagnostic biomarkers and constructed a diagnostic model using the lasso algorithm and multivariate analysis. We developed nomograms to improve clinical application. The ROC curve showed our model had better predictive value for AMD than age and gender, suggesting that our diagnostic model can greatly facilitate the predictive diagnosis of AMD patients. The genes associated with the diagnostic model, including *GABRB3*, *MFF*, and *PROX1*, will be of great benefit to future studies regarding the diagnosis, progression, and treatment of AMD. More studies are needed to further understand the mechanisms of these genes and the role of Tfh cells in AMD.

## Figures and Tables

**Figure 1 diagnostics-13-02732-f001:**
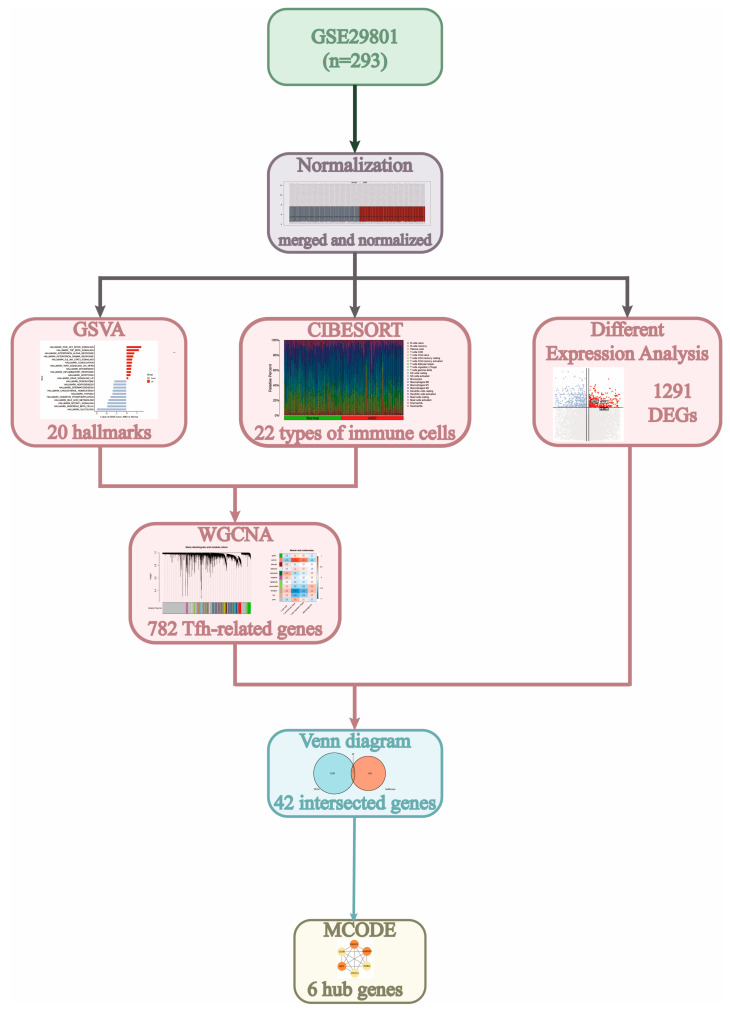
The study process is depicted by using a flow chart. DEG, differently expressed gene; GSVA, gene set variation analysis; WGCNA, weighted gene co-expression network analysis.

**Figure 2 diagnostics-13-02732-f002:**
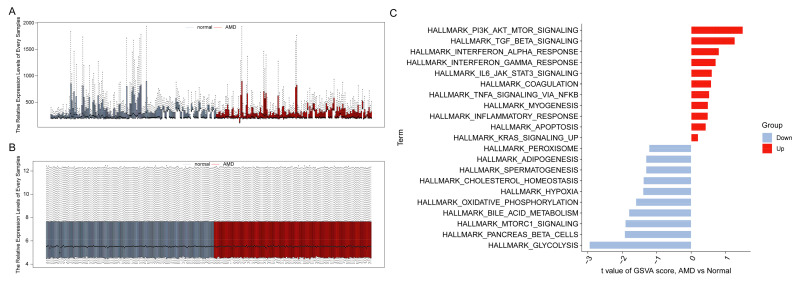
Normalization of data and Gene Set Variation Analysis (GSVA). (**A**) Box plot showing the raw data for all 293 samples. (**B**) Box plot showing the normalized data for all 293 samples. (**C**) GSVA bar plot showing the significantly different Hallmark sets between the AMD and normal groups.

**Figure 3 diagnostics-13-02732-f003:**
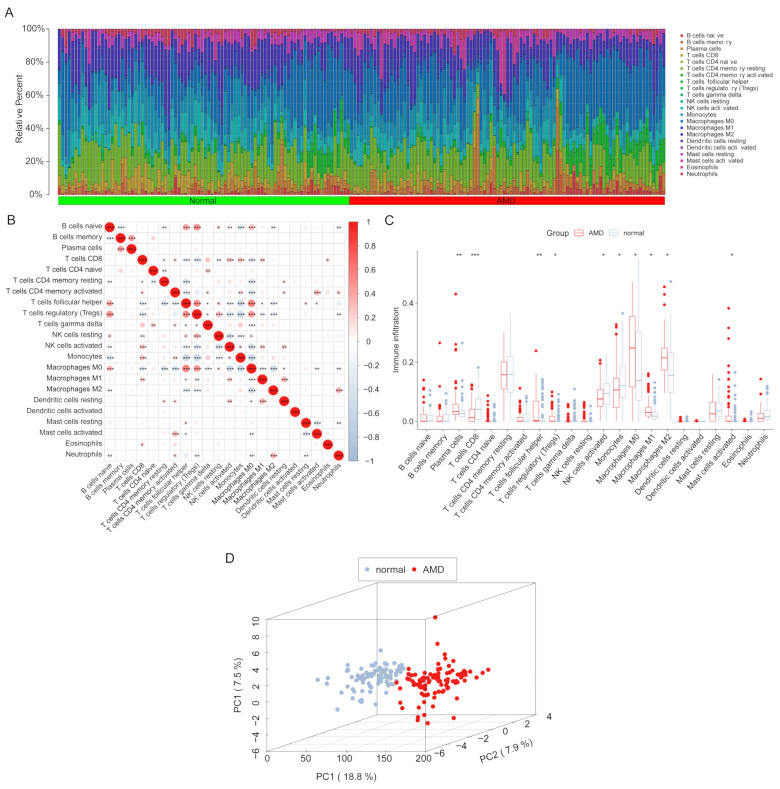
Infiltration levels of 22 immune cells were analyzed among 183 samples. (**A**) Bar-plot displaying the relative proportion levels of all immune cells. (**B**) Correlation heatmap showing the positive and negative relationships among the 22 immune cells (* *p* < 0.05; ** *p* < 0.01, *** *p* < 0.001). (**C**) Boxplot comparing the infiltration levels of immune cells between AMD and normal groups (* *p* < 0.05; ** *p* < 0.01, *** *p* < 0.001). (**D**) Scatterplot of Principal Component Analysis (PCA) indicating significant differences between AMD and normal samples based on immune cell infiltration levels.

**Figure 4 diagnostics-13-02732-f004:**
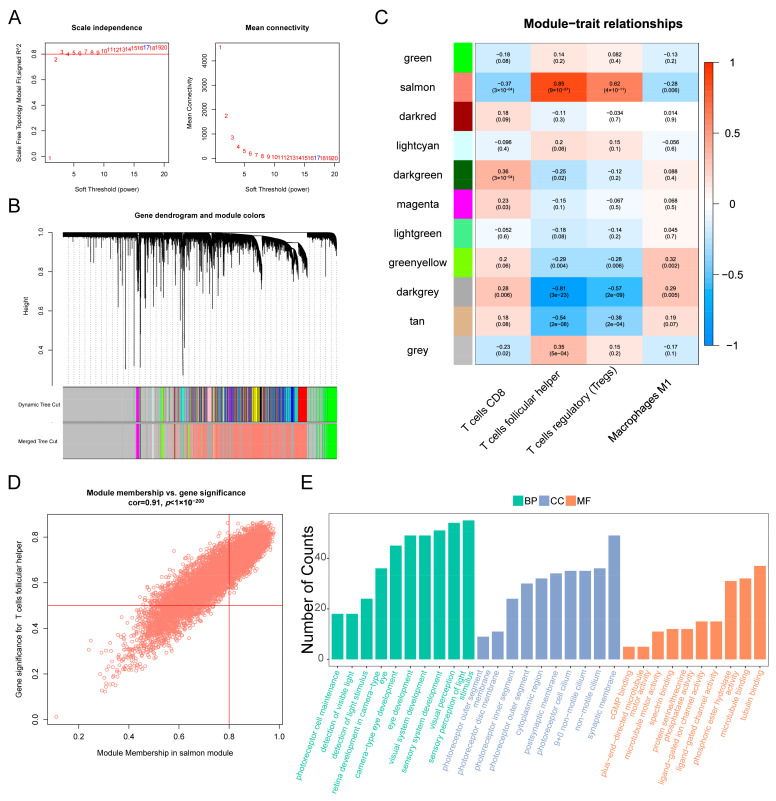
Weighted Gene Correlation Network Analysis (WGCNA) was performed and GO analysis was carried out. (**A**) Scale independence and mean connectivity analysis were performed with soft-power of 17. (**B**) Genes were identified and grouped into different modules using hierarchical clustering method, with different colors representing different modules. (**C**) Correlation heatmap was generated to show the relationship between gene modules and 4 types of immune cells, with red indicating positive correlations and blue indicating negative correlations. (**D**) Scatterplot was generated to show the relationship between significant genes in the salmon module and follicular helper T cells. (**E**) GO analysis was performed using 782 significant genes in the salmon module.

**Figure 5 diagnostics-13-02732-f005:**
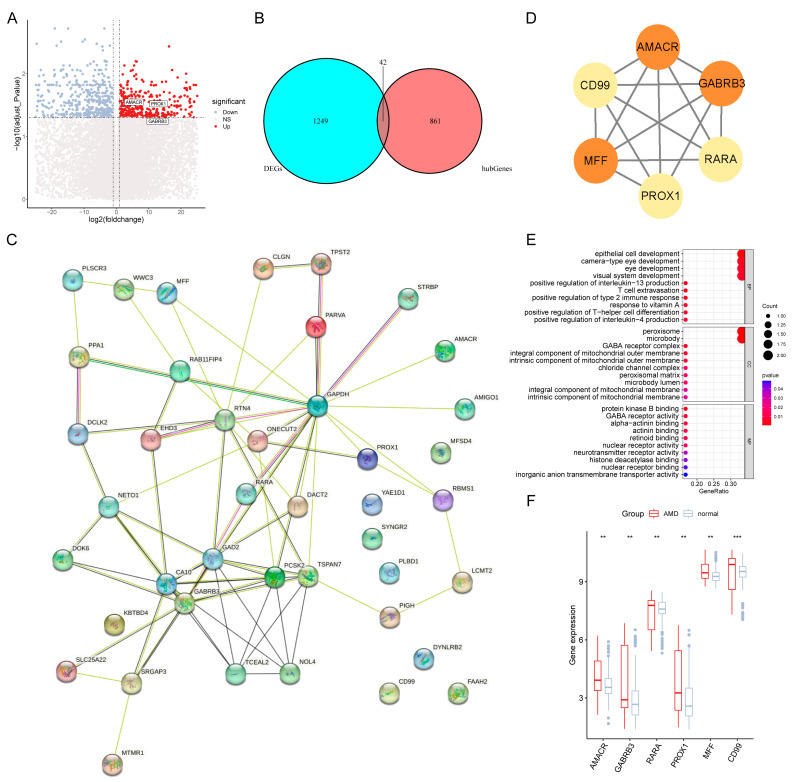
Excavation of Tfh-Related Biological Markers. (**A**) Differential expression analysis among genes in AMDs and controls. Red indicates upregulated genes, blue indicates downregulated genes, and light gray indicates non-significant genes. (**B**) Venn diagram to find the common genes of differentially expressed genes and follicular helper T cell (Tfh)-related genes. (**C**) Protein–protein interaction (PPI) network of 42 intersected genes analyzed from the STRING website. (**D**) PPI network of six hub genes identified by the MCODE function in Cytoscape software (color intensity represents the magnitude of the correlations). (**E**) Gene Ontology analysis based on six potential biomarkers. (**F**) Boxplots for differential analysis of six gene expression levels among AMDs and controls (** *p* < 0.01, *** *p* < 0.001).

**Figure 6 diagnostics-13-02732-f006:**
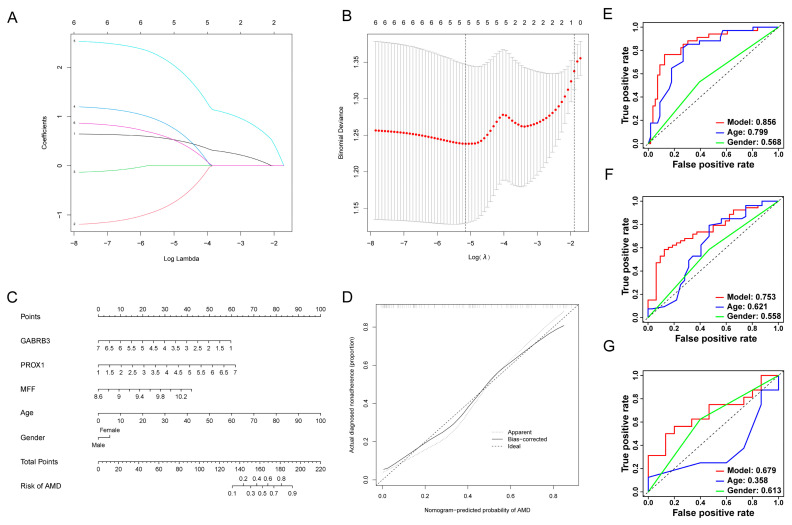
Diagnostic model of age-related macular degeneration. (**A**) *λ*-curve in lasso regression analysis. (**B**) log (*λ*)-curve in lasso regression analysis. (**C**) The nomogram of 3 model-related genes. (**D**) The calibration curve for nomogram. (**E**) ROC curves of diagnostic model, age, and gender in training group. (**F**) ROC curves of diagnostic model, age, and gender in testing group. (**G**) ROC curves of diagnostic model, age, and gender in GSE99248 group.

**Figure 7 diagnostics-13-02732-f007:**
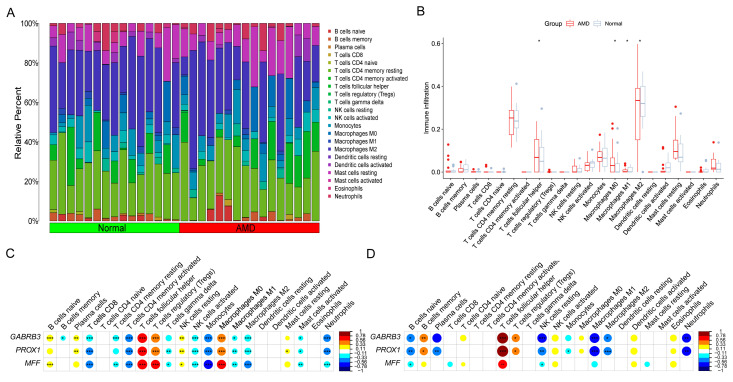
Correlation analysis between model-related risk score and Tfh cells infiltration levels. (**A**) Bar-plot displaying the relative proportion levels of all immune cells. (**B**) Boxplot comparing the infiltration levels of immune cells between AMD and normal groups (* *p* < 0.05; ** *p* < 0.01, *** *p* < 0.001). (**C**) Spearman correlation analysis between three biomarkers and immune cells infiltration levels in GSE29801 group. (**D**) Spearman correlation analysis between three biomarkers and immune cells infiltration levels in GSE99243 group.

**Table 1 diagnostics-13-02732-t001:** Multivariate analysis on 5 genes.

Variable	Coefficient	Odds Ratio (95% CI)	*p*-Value
(Intercept)	−36.214	0 (0~0.004)	0.029
*AMACR*	0.666	1.946 (0.742~5.627)	0.192
*GABRB3*	−1.198	0.302 (0.088~0.930)	0.045
*PROX1*	1.228	3.416 (1.210~10.853)	0.027
*MFF*	2.618	13.712 (1.555~162.893)	0.025
*CD99*	0.848	2.335 (0.554~10.779)	0.258

## Data Availability

The gene expression data and relevant clinical information, GSE29801 and GSE99248, of AMD were downloaded from the Gene Expression Omnibus (GEO) website (http://www.ncbi.nlm.nih.gov/geo/, accessed on 10 January 2023). And the data related with our study have been uploaded in Appendix A. Additional data related to this paper may be requested from the authors.

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
