# Peer review of "Identification of the Immune Landscapes and Follicular Helper T Cell-Related Genes for the Diagnosis of Age-Related Macular Degeneration"

_diagnostics, 2023, doi:10.3390/diagnostics13172732_

Round 1

Reviewer 1 Report

Dear Editor,

Thanks for offering the opportunity to review this paper. The topic of this paper is indeed meaningful. However, there are some problems with this paper that need to be addressed, so I would like to review it again after the author carefully modifies it.

1. Language: On the whole, I can understand what the author is trying to express in this manuscript, but there are some places where the expression is not smooth. Thus the language needs to be polished.

2. I think one of the biggest issues with this article is that the authors (perhaps because they are not ophthalmologists) seem to have a limited understanding of AMD-related concepts. Clinically, age-related macular degeneration (AMD or ARMD) can be divided into two types: atrophic AMD (dry AMD) and exudative AMD (wet AMD). At present, there is no recognized treatment method for dry AMD, although therapies such as complement therapy have shown certain efficacy in clinical trials. However, measures such as Photodynamic therapy and anti VEGF therapy for wet AMD have shown significant benefits for the disease. The author first needs to clarify whether they are studying one type of AMD or a mixture of two types of AMD.Then they need to reorganize the relevant descriptions of this article from beginning to end.

3. I strongly recommend that the author seek the help of an ophthalmologist to revise this paper. E.g., "This stage is known as dry AMD (Line 42)", dry AMD is not a STAGE but a type of AMD.

4. The Figures are too blurry, so it is necessary to improve the resolution.

5. Discussion: the same problem as I have mentioned in Comment 2.

Minor editing of English language required

Author Response

Dear Reviewer:

We would like to express our sincere gratitude to the editors and reviewers for your careful reading, helpful comments, and constructive suggestions, which has significantly improved the presentation of our manuscript.

We have carefully discussed all comments from the reviewers and revised our manuscript accordingly. The manuscript has also been double-checked, and the typos and grammar errors we found have been corrected. Revisions in the manuscript have been marked with black type and a yellow background. In the following section, we summarize our responses to each comment from the reviewers.

Report

Reviewer 1

Thanks for offering the opportunity to review this paper. The topic of this paper is indeed meaningful. However, there are some problems with this paper that need to be addressed, so I would like to review it again after the author carefully modifies it.

  1. Language: On the whole, I can understand what the author is trying to express in this manuscript, but there are some places where the expression is not smooth. Thus, the language needs to be polished.

Response: Thank you very much for your comment. We have revised the manuscript to address the issues you raised pertaining to language fluency and expression. To this end, we have engaged in a thorough language editing process, meticulously refining sentences and reorganizing certain sections to enhance the manuscript's overall readability.

  1. I think one of the biggest issues with this article is that the authors (perhaps because they are not ophthalmologists) seem to have a limited understanding of AMD-related concepts. Clinically, age-related macular degeneration (AMD or ARMD) can be divided into two types: atrophic AMD (dry AMD) and exudative AMD (wet AMD). At present, there is no recognized treatment method for dry AMD, although therapies such as complement therapy have shown certain efficacy in clinical trials. However, measures such as Photodynamic therapy and anti VEGF therapy for wet AMD have shown significant benefits for the disease. The author first needs to clarify whether they are studying one type of AMD or a mixture of two types of AMD. Then they need to reorganize the relevant descriptions of this article from beginning to end.

Response: Your insights into the clinical aspects of age-related macular degeneration (AMD) are highly valuable, and we acknowledge your concern regarding the clarity of our understanding of AMD-related concepts. We are committed to addressing this issue and ensuring that our manuscript accurately reflects the clinical distinctions and nuances of this complex disease. We agree with your assessment that a clear differentiation between atrophic AMD (dry AMD) and exudative AMD (wet AMD) is crucial for the accurate interpretation of our study. In response to your feedback, we will explicitly state the specific type of AMD in our study. This clarification will be prominently placed in both the introduction and methodology sections, ensuring that readers are well-informed about the scope of our research.

Furthermore, we took a comprehensive restructuring of the relevant descriptions throughout the manuscript. By doing so, we aim to enhance the reader's understanding of our research objectives, methodologies, and findings, while also aligning with the clinical classifications of AMD. To address your concerns, we took the following steps: clearly specify we are focusing on a combination of both types in our study. Reorganize the manuscript to present the relevant descriptions of the chosen AMD type(s) in a logical sequence, ensuring a coherent and comprehensive narrative. We are genuinely grateful for your insightful guidance, which will undoubtedly contribute to the accuracy and clinical relevance of our manuscript. We are committed to making these revisions diligently and promptly to meet the high standards set by Diagnostics.

  1. I strongly recommend that the author seek the help of an ophthalmologist to revise this paper. E.g., "This stage is known as dry AMD (Line 42)", dry AMD is not a STAGE but a type of AMD.

Response: Thank you for your dedicated review and insightful recommendations.

Your point regarding the classification of dry AMD as a type rather than a stage is indeed valid, and I wholeheartedly acknowledge the distinction.  We fully understand the importance of using precise terminology, especially in a specialized field such as ophthalmology.

To address this concern and ensure the accurate representation of AMD-related concepts, we have taken the following steps: iinvited an ophthalmologist with expertise in AMD to thoroughly review and provide guidance on the clinical accuracy and terminology used in the manuscript. Revise all instances where the terms "stage" and "type" have been misused or interchanged to accurately reflect the clinical classifications of AMD. By involving an ophthalmologist in the revision process, we are committed to achieving the highest level of accuracy and clinical relevance in our manuscript. We appreciate your suggestion and recognize its pivotal role in elevating the quality of our work.

  1. The Figures are too blurry, so it is necessary to improve the resolution.

Response: Thank you for your valuable review of our manuscript. We apologize for the inconvenience caused by the blurry figures and are committed to rectifying this issue. We understand the critical role that clear and high-resolution figures play in enhancing the reader's understanding of our research findings.

To address the concern of blurry figures, we will undertake the following measures:

Image Enhancement: We utilized appropriate image editing software to enhance the resolution and clarity of all figures. This adjustment improves parameters such as sharpness, contrast and brightness, while ensuring that data integrity and presentation remain intact.

Higher Resolution Images: If the originally generated figures have a lower resolution, we revised them using higher resolution source data to ensure optimal clarity.In cases where the figures were originally generated at a lower resolution, we will recreate these figures using higher resolution source data to ensure the best possible view.Figure Captions: We also revised the figure captions to ensure their accuracyfor the clear understanding of the readers.

By implementing these steps, we aimed to significantly improve the quality of the figures in the manuscript and provided readers with a clearer representation of our research.

  1. Discussion: the same problem as I have mentioned in Comment 2.

Response: Thank you very much for your suggestion. I understand your concern about the issue highlighted in Comment 2, which extends to the Discussion section of the manuscript as well. To specifically address this issue in the Discussion section, we have taken the following actions:

Review and Alignment: We revised the Discussion section to ensure that concepts, ideas, and conclusions are presented in a coherent and logically structured manner. We paid particular attention to any instances where the expression may not be smooth or may lead to confusion.

Terminology Clarification: We have carefully proofread the article to ensure correctness and uniformity in the precise use of relevant terms throughout the discussion section. This will help avoid ambiguity and contribute to a more accurate presentation of our findings and interpretations.

Thank you again for your valuable and constructive comments which helped us a lot in revising this article.

Sincerely,

Yao Yang

Reviewer 2 Report

In the paper “Identification of the Immune Landscapes and Follicular Helper T Cell-related Genes for the Diagnosis of Age-related Macular Degeneration”, Yang et al. studied the role of Follicular helper T (Tfh) cells in the development of Age-related Macular Degeneration (AMD) through different bioinformatics analysis techniques. They showed that the AMD samples has a higher level of different immune cell infiltration compared to the normal samples. Through the analyses, the authors also selected three diagnostic biomarker genes for AMD which they found to be associated with the Tfh cells. The paper is interesting and informative. For publication in Diagnostics, my specific comments are as follows:

1.       Fig 2A and 2B. The label of the y-axes is missing.

2.       The stacked bar plot representation of the immune cell infiltration percentages presented in Fig 3A is a bit hard to interpret due to the large number of samples and not-so-high figure quality. It would be better if the authors use a different representation for that data.

3.       The source and the type (e.g., microarray or RNA-seq) of the immune cell gene expression data are not mentioned in the manuscript.

4.       From the PCA plot of Fig. 3D, it looks like the PC1 itself is able to separate the normal samples from the AMD samples pretty well. In that case, it would be very interesting to see what genes are responsible for the separation along the PC1 axis. The genes can be selected based on their contribution from the corresponding eigenvector. Are those genes also come up when the downstream DEG analysis and biomarker selection steps are performed? For the PCA plot, I would also suggest adding the percentage of variance explained by each PCs to the figure.

5.       The authors showed that their nomogram model with three genes has improved AMD diagnostic potential more than the individual clinical properties. However, it is possible that adding the clinical properties along with the three gene features will be able to further increase the performance of the model. This would be very interesting if the clinical features could be combined with genomic features for better diagnostic potential.

6.       In Fig. 7C and 7D, a significant number of samples, which have widely varying risk scores, exhibit no Tfh cell infiltration. It is highly unlikely that the Tfh cell infiltration percentage itself can predict the risk score.

7.       The module trait relationship plot in Fig 4C shows that the regulatory T cell also has high enrichment for the salmon module, although not to the level of Tfh. Is it possible that the biomarkers identified are also associated with Treg cells?

8.       The resolution of the figures are needed to be improved.

I couldn't find any problem with their English language.

Author Response

Dear Reviewer:

We would like to express our sincere gratitude to the editors and reviewers for your careful reading, helpful comments, and constructive suggestions, which has significantly improved the presentation of our manuscript.

We have carefully discussed all comments from the reviewers and revised our manuscript accordingly. The manuscript has also been double-checked, and the typos and grammar errors we found have been corrected. Revisions in the manuscript have been marked with black type and a yellow background. In the following section, we summarize our responses to each comment from the reviewers.

Report

Reviewer #2:

In the paper “Identification of the Immune Landscapes and Follicular Helper T Cell-related Genes for the Diagnosis of Age-related Macular Degeneration”, Yang et al. studied the role of Follicular helper T (Tfh) cells in the development of Age-related Macular Degeneration (AMD) through different bioinformatics analysis techniques. They showed that the AMD samples has a higher level of different immune cell infiltration compared to the normal samples. Through the analyses, the authors also selected three diagnostic biomarker genes for AMD which they found to be associated with the Tfh cells. The paper is interesting and informative. For publication in Diagnostics, my specific comments are as follows.

  1. Fig 2A and 2B. The label of the y-axes is missing.

Response: Thank you very much for your comment. We added the correct labels in Figure 2A and 2B at line 185, Page 6.

  1. The stacked bar plot representation of the immune cell infiltration percentages presented in Fig 3A is a bit hard to interpret due to the large number of samples and not-so-high figure quality. It would be better if the authors use a different representation for that data.

Response: Thank you very much for your suggestion. This is indeed a very important issue for the better understanding of the manuscript by the readers. We searched the literature for study similar to ours and found that most researchers chose stack histograms to visualize the proportion of 22 immune cells in each sample. However, due to the low quality and resolution of the images, readers cannot clearly distinguish each set of data, too. Therefore, we increased the width of each column to improve the quanlity of the images at line 202, page 7.  If you have any other form of recommended representation for that data, please inform us.  We will change the data as you suggested. Thanks.

  1. The source and the type (e.g., microarray or RNA-seq) of the immune cell gene expression data are not mentioned in the manuscript.

Response: Thank you very much for your careful check. Your misinterpretation is due to incomplete explanation of our methodology. The immune cell infiltration level files in this study were predicted by the CIBERSORT algorithm of the IBOR package in R software, and the corresponding immune cell gene expression data were not downloaded from other databases. We have added a corresponding note in the subsection "2.3 CIBERSORT analysis" in the "Materials and Methods" section at line 121, page 3.

  1. From the PCA plot of Fig. 3D, it looks like the PC1 itself is able to separate the normal samples from the AMD samples pretty well. In that case, it would be very interesting to see what genes are responsible for the separation along the PC1 axis. The genes can be selected based on their contribution from the corresponding eigenvector. Are those genes also come up when the downstream DEG analysis and biomarker selection steps are performed? For the PCA plot, I would also suggest adding the percentage of variance explained by each PCs to the figure.

Response: We apologize for any misunderstanding due to our incomplete presentation. Figure 3D was plotted based on the degree of infiltration of 22 immune cells to show that there is some difference in the degree of immune cell infiltration between the control group and the AMD group, rather than the PCA plots of the samples based on the expression levels of the genes. Also, as per your suggestion, we have added the corresponding percentage of variance to each axis of the PCA plot at line 202, page 7.

  1. The authors showed that their nomogram model with three genes has improved AMD diagnostic potential more than the individual clinical properties. However, it is possible that adding the clinical properties along with the three gene features will be able to further increase the performance of the model. This would be very interesting if the clinical features could be combined with genomic features for better diagnostic potential.

Response: Thank you very much. According to your suggestion, we added the clinical traits of age and gender to the Nomogram model to improve the prediction accuracy of our model, and redrawn the relevant pictures in Figure 6B-G at line 227, page 10, and uploaded the revised pictures together with the manuscript.

  1. In Fig. 7C and 7D, a significant number of samples, which have widely varying risk scores, exhibit no Tfh cell infiltration. It is highly unlikely that the Tfh cell infiltration percentage itself can predict the risk score.

Response:  Thank you for your comments. We have noticed that Tfh cells are a relatively small number of immune cells among the 22 types of immune cells. In addition, since the CIBERSORT algorithm is to calculate the relative proportion of 22 types of immune cells in each sample, it inevitably leads to the fact that the Tfh infiltration degree of some samples is close to zero compared with other immune cells. At the same time, we first drew the correlation dot-plots between the newly constructed risk score with clinical factors and Tfh cells, and the results showed that there was a very weak correlation between the them. The relevant pictures were uploaded as a material not for publication, and the file name was "Correlation between Risk-Score and Tfh". In addition, given that the main purpose of this study was to identify biomarkers related to AMD, we removed Figure 7C-7E from the original figure and added two correlation heatmaps of GSE29801 and GSE99248 between the three markers and 22 immune cells, which can be seen in the revised Figure 7C,D at line 300, page 11. These is worth noting that this coincides with your Comment 7.

  1. The module trait relationship plot in Fig 4C shows that the regulatory T cell also has high enrichment for the salmon module, although not to the level of Tfh. Is it possible that the biomarkers identified are also associated with Treg cells?

Response: Thank you for your suggestion. According to your suggestion, we further analyzed the correlation between the screened 3 biomarkers and the infiltration degree of 22 kinds of immune cells, and drew the correlation heatmaps in Figure 7C and 7D. The results show: in the GSE29801 and GSE99243 datasets, all three biomarkers showed a strong positive correlation with Tfh cells but a weak correlation with Treg cells. These revealed that although there have some associations between three biomarkers, but they are mainly correlated with Tfh cells. The relevant words were revised and added at line 294-299, page 11.

  1. The resolution of the figures are needed to be improved.

Responds: Thank you very much for your suggestion. Based on your suggestions, we have reprocessed the images in the manuscript to increase the resolution.

Thank you once again for your dedication to advancing scholarly excellence in our field.

Sincerely,

Yao Yang

Round 2

Reviewer 1 Report

Can be accepted now.

Minor editing of English language required

Author Response

Dear Reviewer:

    Thank you so much for your reviewing! We deeply appreciate your recognition of our research work. Special thanks to you for your good comments.

    Thank you once again for your dedication to advancing scholarly excellence in our field.

Sincerely,

Yao Yang

Reviewer 2 Report

The quality of Fig. 6 E, F, and G still needs improvement. The AUC scores are unreadable.

I have no problem with the quality of the English language.

Author Response

Dear Reviewer:

    We would like to express our sincere gratitude for your careful reading, helpful comments, and constructive suggestions, which has significantly improved the presentation of our manuscript.

    We have carefully discussed your comments and revised our manuscript accordingly. In the following section, we summarize our responses to your comments.

Comments: The quality of Fig. 6E, F and G still needs improvement The AUC scores are unreadable.

Responses: Thank you so much for your reviewing! To enhance the clarity and readability of the text in Figure. 6E-G., we widened the corresponding line segment and increased the font sizes. The revised image has been updated in our manuscript by using the “Track Changes” function, and uploaded to the submission system.

    Thank you once again for your dedication to advancing scholarly excellence in our field.

Sincerely,

Yao Yang